# Chemical Synthesis, Safety and Efficacy of Antihypertensive Candidate Drug 221s (2,9)

**DOI:** 10.3390/molecules28134975

**Published:** 2023-06-25

**Authors:** Bei Qin, Lili Yu, Rong Wang, Yimei Tang, Yunmei Chen, Nana Wang, Yixin Zhang, Xiong Tan, Kuan Yang, Bo Zhang, Maofang He, Yuzhen Zhang, Yaqi Hu

**Affiliations:** 1Xi’an Key Laboratory of Multi Synergistic Antihypertensive Innovative Drug Development, Xi’an Medical University, Xi’an 710021, China; yulili@xiyi.edu.cn (L.Y.); wangrong@xiyi.edu.cn (R.W.); tangym@xiyi.edu.cn (Y.T.); yunmeichen@xiyi.edu.cn (Y.C.); wangnana@xiyi.edu.cn (N.W.); lzhangyixin@163.com (Y.Z.); tanxiong0408@163.com (X.T.); yangkuan@xiyi.edu.cn (K.Y.); zhangbo@xiyi.edu.cn (B.Z.); hemaofang@xiyi.edu.cn (M.H.); zhangyuzhen@xiyi.eud.cn (Y.Z.); qiqihu1988@xiyi.edu.cn (Y.H.); 2Institute of Drug Research, Xi’an Medical University, Xi’an 710021, China; 3College of Pharmacy, Xi’an Medical University, Xi’an 710021, China; 4School of Pharmacy, Chengdu Medical College, Chengdu 610500, China

**Keywords:** ACEI, tanshinol, borneol, antihypertensive, RAAS

## Abstract

Hypertension is the main risk factor of cardiovascular and cerebrovascular diseases. In this paper, a novel compound known as 221s (2,9), which includes tanshinol, borneol and a mother nucleus of ACEI, was synthesized by condensation esterification, deprotection, amidation, deprotection, and amidation, with borneol as the initial raw material, using the strategy of combinatorial molecular chemistry. The structure of the compound was confirmed by ^1^H NMR, ^13^C NMR, and high-resolution mass spectrometry, with a purity of more than 99.5%. The compound 221s (2,9) can significantly reduce the systolic and diastolic blood pressure of SHR rats by about 50 mmHg and 35 mmHg after 4 weeks of administration. The antihypertensive effect of 221s (2,9) is equivalent to that of captopril. The use of 221s (2,9) can reduce the content of Ren, Ang II and ACE in the serum of SHR rats, inhibit the RAAS and enhance the vascular endothelial function by upregulating the level of NO. Pathological studies in this area have shown that high dosage of 221s (2,9) can notably protect myocardial fibrosis in rats and reduce the degeneration and necrosis of myocardial fibers, inflammatory cell infiltration, and proliferation of fibrous tissue in the heart of rat. Therefore, the existing work provided a foundation for preclinical research and follow-up clinical research of 221s (2,9) as a new drug.

## 1. Introduction

Hypertension, characterized by continuous systemic arterial hypertension, is the leading risk factor of cardiovascular and cerebrovascular diseases in the world [1], affecting over a quarter of adults. To date, the treatment of hypertension has been divided into drug treatment [2] and non-drug treatment [3]. Drug treatment is the most effective means [3], contains diuretic [4,5,6], CCB [7], ACEI [8], ARB [9], β receptor blockers [10] and α receptor blocker [11]. The renin–angiotensin–aldosterone system (RAAS) was over-activated in the proceeding of hypertension, resulting in vasoconstriction, vascular and cardiac hypertrophy and fibrosis and inducing arteriosclerosis and cardiac dysfunction. These effects led to complications of cardiovascular and renal systems [12]. Therefore, RAAS has become an important anti-tumor strategy for the treatment of hypertension.

Angiotensin-converting enzyme inhibitors (ACEI) can significantly reduce the concentration of Ang II in plasma and inhibit the pressor effect of exogenous Ang I by directly inhibiting ACE and the production of angiotensin II in the RAAS [8]. Additionally, ACEI possess the function of expanding coronary artery [13], increasing myocardial blood supply [14,15] and improving renal blood flow [16]. Therefore, ACEI has become the commonly used drug in clinics at present. However, ACEI have various adverse effects such as an irritant dry cough [17,18,19], vascular edema [20], hyperkalemia [21,22], renal dysfunction [23] and hypotension [24].

The RAAS is composed of renin (Ren), angiotensin II (Ang-II) and aldosterone (ALD). Ren, a pre-hormone synthesized by juxtaglomerular cell, can decompose angiotensinogen into decapeptide angiotensin I (Ang I) after activation. ATI can be converted into angiotensin II (Ang II) by angiotensin-converting enzyme (ACE) in pulmonary capillaries, endothelial cells and renal epithelial cells [25]. Ang II is an effective vasoconstrictor that can increase total peripheral resistance and mean arterial pressure and promote the release of ALD from the adrenal cortex. ALD enhances the retention of sodium and water in the kidney, increases venous reflux and cardiac output, and then increases total peripheral resistance and venous reflux. In addition, Ang II exerts the function of vasoconstriction through angiotensin type 1 (AT-1) and angiotensin type 2 (AT-2) receptors.

It is widely accepted that the residue (carbon end) of proline (Pro), binding to the active center of ACE, is the essential group of ACEI that exerts antihypertensive activity [26]. As an endogenous substance [27], tanshinol possesses the pharmacological effects of promoting angiogenesis, expanding coronary artery [28], improving myocardial injury [29], and protecting the organs of blood vessels [30], liver [31] and heart [32]. Additionally, the catechol and lactic acid structure in tanshinol can coordinate with the zinc ion in the active center of ACE enzyme. This can increase the binding ability of active molecules and ACE, thus improving the inhibition rate of drug molecules on ACE. Drugs containing tanshinol are still widely used in the clinic, such as danshen tablets, compound danshen dripping pills, etc. Borneol, also known as D-borneol, has been shown to assist drugs to penetrate the brain blood barrier and increase the quantity of drug molecules reaching the brain [33].

Based on the research strategy into multi-molecule synergy, our lab synthesized a series of 221s (2,9) drug active molecules by designing a combined structure of tanshinol, borneol and the mother nucleus of ACEI, and by screening out a molecule with good ACE inhibitory activity. The main objective of this study was to scale up the synthesis and quality evaluation of candidate compound 221s (2,9), with a focus on the acute toxicity, antihypertensive efficacy, and preliminary safety of the compound. This study will promote the preclinical study of 221s (2,9) as a candidate compound.

## 2. Results and Discussion

### 2.1. Synthesis of 221s (2,9)

An ACEI antihypertensive drug with tanshinol, borneol and the mother nucleus of ACEI was designed and synthesized based on drug molecular structure design. Proline and alanine in the structure constitute the mother nucleus of ACEI. Danshensu and borneol are added to improve the synergistic antihypertensive effect of drugs, organ protection, and the ability of drugs to pass through the blood–brain barrier. The molecule contains a mother nucleus composed of proline and alanine, and the two amino acids and danshensu are connected by two amide bonds similar to dipeptide. Borneol is linked to amino acids by ester bonds. Therefore, the conventional synthetic methods of peptides and esters were mainly referred to in the synthesis of this molecule. The first step is the synthesis of Boc-l-proline bornyl ester. As reported in the literature, the most commonly used methods for the synthesis of proline ester reported are direct esterification [34] and condensation agent esterification [35]. The acid condition of direct esterification can easily lead to the hydrolysis of ester structure, while the condensation reaction under the action of EDCI and DMAP is milder, the yield is higher and the isomerization ratio is lower, meaning it is therefore the preferred method of routing. Two group processes for the deprotection of Boc group processes are involved in the synthetic route. Generally, the most common reaction condition in the literature is the trifluoroacetic acid dichloromethane solution [36]. However, due to the presence of borneol esters in the reaction substrate structure, the strong acidic condition will promote the hydrolysis of borneol esters, seriously affecting the yield of the reaction, and increasing the difficulty of purification. It is also reported that the Boc protection group can be removed by HCl and TBDMS-OTf [37], but that ester bond hydrolysis is found in these methods. Therefore, a relatively mild 85% phosphoric acid solution was selected to achieve the removal of the protective group in order to minimize the hydrolysis problem. There are phenol hydroxyl and alcohol hydroxyl groups in the structure of Danshensu that need to be protected by hydroxyl groups and then coupled with an amino acid during the amidation reaction [38]. Our literature search found that it is rare to realize the connection between Danshensu and amine through a one-step condensation reaction. According to the characteristics of the reaction, the direct condensation reaction of Danshensu and amine was optimized, and the direct amidation of Danshensu and compound **4** was realized. According to the characteristics of the reaction, the research team optimized the direct condensation reaction of Danshensu and amine, finally realizing the direct amidation of Danshensu and compound **4**, while they improved the feasibility of reaction amplification.

### 2.2. Analysis of 221s (2,9)

The objective samples were analyzed and calibrated using HPLC methods with gradient elution. The factor of spectral purity for the main maximum peak was confirmed to be 998 on the DAD detector, which showed that the main peak only stood for a single component (Appendix A). The purity was confirmed by HPLC to be more than 99.87%, which was calculated by the area normalization method, and the total impurity content was 0.13% (Appendix A). On the basis of chromatographic purity, drying shrinkage (0.0059%) and residue on ignition (0.110%), the purity of objective sample was calculated to be 99.75%. With the thermal analysis method, the purity was determined to be 99.51%. The above-mentioned methods illustrated that the purity of objective sample was more than 99.5% (Appendix A).

### 2.3. Acute Toxicity Test of 221s (2,9)

The LD50 of 221s (2,9) was measured according to technical guidelines for the acute toxicity test of chemical drugs. It is impossible to measure the LD50 of 221s (2,9) as no dead rats were observed. Therefore, a maximum tolerance dose of 221s (2,9) was observed. A total of 24 mice were randomly divided into 2 groups (12 mice/group): (1) control group, given CMC-Na; and (2) 221s (2,9), given oral administration of 221s (2,9) (3000 mg/kg). The survival condition and poisoning situation of rats were measured for 7 days after oral administration of the doses. The results showed that all mice were alive and that no pathological change was observed in the tissues of mice. Therefore, the maximum tolerance dose of 221s (2,9) was 3000 mg/kg.

### 2.4. The Effect of 221s (2,9) on Blood Pressure, Heart Rate and Body Weight of Rats 

As shown in Figure 1a,b, the blood pressure (BP) in the control was stable and the systolic blood pressure (SBP) and diastolic blood pressure (DBP) of SHR model group increased with time. However, the treatment of 221s (2,9) and captopril alleviated the enhancement in the SBP and DBP of the SHR model group (*p* < 0.05). In addition, the 221s (2,9) group showed a decrease in SBP and DBP by 50 mmHg and 35 mmHg, respectively. The antihypertensive effect of the 221s (2,9) group exhibited no significant difference compared to the Captopril/SHR group. As illustrated in Figure 1c,d, 221s (2,9) showed no effect on the heart rate and body weight of rats. The above results indicated that 221s (2,9) has equal or even better effects on hypertension than captopril. 

### 2.5. Measurement of REN, AngII, ALD, and ACE in Serum

It is widely accepted that the RAAS plays a key role in the development of hypertension and that Ang II, a strong vasoconstrictor, is the main active substance of the RAAS. ACE can catalyze the conversion of angiotensin I into Ang II via the cleavage of the C-terminal dipeptide. This leads to the activation of AT-1R and AT-2R, lowering the level of aldosterone and vasopressin. ACEI decreased the SBP and DBP of the body in a dose-dependent manner [39]. As shown in Figure 2, a high dose and low dose of 221s (2,9) can decrease the level of REN and Ang II in the serum of rats, and inhibit the concentration of ACE, indicating that 221s (2,9) may depress the RAAS.

### 2.6. Measurement of NO and ET-1 in Serum

It is reported that endothelial dysfunction is the main pathogenesis of hypertension. Endothelin-1 (ET-1) is a vasoconstrictor secreted by endothelial cells that acts as the natural counterpart to the vasodilator nitric oxide (NO) [40]. The level of both can reflect the vascular endothelial function. The results showed that (Figure 3) 221s (2,9) had no obvious effect on ET-1 level; however, 221s (2,9) upregulated the level of NO in rat serum, suggesting that 221s (2,9) could improve vascular endothelial function.

### 2.7. HE Staining

HE staining was used to assess the effect of 221s (2,9) on cardiac histopathological changes in mice. Chromatin and ribophagy was stained as violet-blue and the ingredients of cytoplasm and extracellular matrix were stained red. The HE staining result suggests (Figure 4) that the myocardial cells in the control group had clear cross-striation, uniform staining, and an orderly arrangement. However, the myocardial tissue in the SHR model group was seriously damaged, had uneven staining, blurred cross-striation, disordered arrangement, a large amount of cellular infiltration, and flaky necrosis, and its cell nuclei were smaller. The compound 221s (2,9) could relieve the pathological change in SHR model group in a dose-dependent manner.

### 2.8. Masson Staining

Masson’s staining, which uses Weigert’s Hematoxylin, Biebrich scarlet-acid fuschin solution, and Aniline blue for staining, is a histological staining method used to selectively stain collagen, collagen fibers, fibrin, muscles, and erythrocytes. Image-Pro Plus 6.0 software was used to measure the relative area of fibrous tissue. Masson staining results (Figure 5) showed that the relative area of the fibrous tissue of the control group, SHR model group, captopril group, high dose of 221s (2,9) group and low dose of 221s (2,9) group was 0.31 ± 0.15, 3.86 ± 3.15, 2.63 ± 2.13, 1.11 ± 0.23 and 2.74 ± 1.35, respectively, suggesting that 221s (2,9) could relieve myocardial fibrosis in a dose-dependent manner.

This section may be divided by subheadings. It should provide a concise and precise description of the experimental results, their interpretation, as well as the experimental conclusions that can be drawn.

## 3. Experimental Section

### 3.1. Reagent

1-Hydroxybenzotriazole(HOBT), 1-Ethyl-(3-dimethylaminopropyl) carbodiimide hydrochloride (EDCI) and N-Boc-L-Alanine were purchased from Shanghai McLean Biochemical Technology Co., Ltd. (Shanghai, China); 4-Dimethylaminopyridine (DMAP) was purchased from Suzhou Fritz Testing Technology Co., Ltd. (Suzhou, China); tanshinol and Borneol were provided by Northwest University; Captopril tablets (25 mg/tablet) were provided by Shanxi Jinhuahuixing Pharmaceutical Co., Ltd. (Yuncheng, China) (GYZZ: H19993357; batch number: 20201103); and ELISA test kits of REN, AngII, ALD, NO and ET-1 were purchased from Shanghai Enzymes Biotechnology Co., Ltd. (Shanghai, China).

### 3.2. Experimental Animal

Male SPF SHRs (spontaneous hypertension rats), 8 weeks old (weight 180–200 g), were provided by Sperford Biotechnology Co., Ltd. (Beijing, China) (SCXK (Beijing) 2019-0010). Male SPF grade WKY (Wistar kyoto) rats, 8 weeks old (weight 180–200 g), were provided by Beijing Weitong Lihua Experimental Animal Technology Co., Ltd. (Beijing, China) (SCXK (Beijing) 2021-0006). The animal protocol was approved by the Animal Ethics Committee of Xi’an Medical College.

### 3.3. Experimental Apparatus

The following equipment and techniques were used: high-performance liquid chromatography (Aligent, Santa Clara, CA, USA); Q600SDT thermogravimetry and Q1000 differential scanning calorimetry synchronous tester (TA Company, New Castle, DE, USA); 20,919 dual-channel non-invasive blood pressure measuring instrument (Kent Scientific, Torrington, CT, USA); Model 1510 automatic full-wavelength enzyme marker (Thermo Fisher Scientific, Waltham, MA, USA); AX224ZH analytical balance (Aarhus Instruments Co., Ltd. Shanghai, China); T10 handheld homogenizer (IKA, Stauffen, Germany); Leica-2016 rotary microtome (Leica, Wetzlar, Germany); and Eclipse Ci microscope (Nikon, Tokyo, Japan).

### 3.4. Synthesis of 221s (2,9)

According to the synthetic route of Figure 1, 221s (2,9) (Tanshinol acyl-l-alanyl-l-proline bornyl ester) was prepared from borneol, N-Boc-l-proline, N-Boc-l-alanine, and tanshinol via condensation, esterification, deprotection and an amidation reaction.

#### 3.4.1. Synthesis of Compound **1**

EDCI (55.9 g, 0.292 mol) and DMAP (3.6 g, 0.029 mol) were added to 630 mL of DCM after borneol (45.0 g, 0.292 mol) and N-Boc-L-proline (63.8 g, 0.297 mol) were dissolved and stirred overnight at room temperature. The reaction products were washed in 5% sodium bicarbonate aqueous solution (450 mL × 3), 10% HCl aqueous solution (450 mL × 3) and water (450 mL) after the reaction was completed by detection of TLC. Then, they were dried with anhydrous sodium sulfate. A total of 85.6 g (0.244 mol) of milky white solid (compound **1**) with a yield of 83.6% was obtained after decompressing and the concentrated extract was dried in vacuum conditions.

^1^H NMR (400 MHz, CDCl_3_) δ 4.95 (dd, *J* = 30.4, 9.7 Hz, 1H), 4.30 (ddd, *J* = 36.4, 8.6, 2.8 Hz, 1H), 3.62–3.34 (m, 2H), 2.37 (ddt, *J* = 13.8, 8.8, 4.5 Hz, 1H), 2.28–2.14 (m, 1H), 1.93 (tdd, *J* = 13.5, 11.5, 9.2, 6.2 Hz, 4H), 1.75 (td, *J* = 8.0, 4.3 Hz, 1H), 1.68 (dt, *J* = 8.3, 4.6 Hz, 1H), 1.44 (d, *J* = 14.3 Hz, 9H), 1.33–1.20 (m, 2H), 0.99 (td, *J* = 14.8, 13.9, 3.4 Hz, 1H), 0.91 (d, *J* = 6.2 Hz, 3H), 0.87 (d, *J* = 5.0 Hz, 3H), 0.83 (d, *J* = 6.2 Hz, 3H).

^13^C NMR (101 MHz, CDCl_3_) δ 173.29, 153.75, 80.27, 79.93, 79.72, 79.43, 59.34, 59.11, 48.81, 47.80, 46.42, 46.24, 44.90, 44.87, 36.78, 36.53, 31.03, 30.10, 28.43, 28.35, 28.04, 27.98, 27.13, 27.10, 24.21, 23.45, 19.62, 18.78, 13.49, 13.34.

Calculate *m*/*z* from C_20_H_33_NO_4_: 351.24, found: 374.22919 ([C_20_H_33_NO_4_ + Na]^+^).

#### 3.4.2. Synthesis of Compound **2**

Compound **1** (85.6 g, 0.244 mol) was dissolved in DCM (128 mL). Then, 85% phosphoric acid solution was added slowly and the solution was stirred overnight. Then, 1800 mL of 10% sodium carbonate aqueous solution was added to the reaction system after the reaction was completed by the detection of TLC. This was followed by stirring until no bubbles were observed. A total of 61.2 g (0.243 mol) of slight yellow oil compound with a yield of 99.6% was obtained after the compound was extracted by DCM (500 mL × 3), dried with anhydrous sodium sulfate, and decompressed.

#### 3.4.3. Synthesis of Compound **3**

Compound **2** (61.2 g, 0.243 mol) was dissolved in a DMF and DCM mixture solvent (700 mL, V:V = 1:1). Then, N-Boc-l-Alanine (50.7 g, 0.268 mol), EDCI (140 g, 0.730 mol) and DMAP (29.7 g, 0.243 mol) were added in sequence, stirring overnight. Then, 500 mL DCM were added to the reaction system, followed by washing with 5% sodium bicarbonate aqueous solution (600 mL), extracting with DCM (500 mL × 3). The organic phases were collected and washed by 5% sodium bicarbonate aqueous solution (600 mL × 2), 10% HCl aqueous solution (600 mL × 3) and water (450 mL), which was followed by drying with anhydrous sodium sulfate, concentrating and vacuum-drying. Finally, 81.2 g of light yellow oil (compound 3, 0.192 mol) was obtained, with a yield of 79.0%.

^1^H NMR (400 MHz, CDCl_3_) δ 5.40 (d, *J* = 8.3 Hz, 1H), 5.00–4.89 (m, 1H), 4.56 (dd, *J* = 8.5, 3.9 Hz, 1H), 4.47 (t, *J* = 7.6 Hz, 1H), 3.71 (t, *J* = 8.1 Hz, 1H), 3.66–3.53 (m, 1H), 2.35 (ddd, *J* = 14.3, 10.1, 4.2 Hz, 1H), 2.28–2.19 (m, 1H), 2.11–1.95 (m, 3H), 1.88 (ddd, *J* = 13.9, 9.7, 4.2 Hz, 1H), 1.74 (tt, *J* = 7.7, 3.6 Hz, 1H), 1.68 (q, *J* = 3.9 Hz, 1H), 1.43 (d, *J* = 2.9 Hz, 9H), 1.36 (d, *J* = 7.0 Hz, 3H), 1.27 (dt, *J* = 9.5, 3.1 Hz, 2H), 1.03 (dd, *J* = 13.9, 3.3 Hz, 1H), 0.88 (dd, *J* = 10.7, 2.7 Hz, 6H), 0.81 (d, *J* = 2.8 Hz, 3H).

^13^C NMR (101 MHz, CDCl_3_) δ 172.01, 171.40, 155.14, 80.51, 79.39, 59.03, 48.90, 47.90, 47.69, 46.72, 44.84, 36.49, 29.16, 28.33, 27.96, 27.11, 24.86, 19.63, 18.75, 18.46, 13.45.

Calculate *m*/*z* from C_23_H_38_N_2_O_5_: 422.28, found: 423.28430 ([C_23_H_38_N_2_O_5_ + H]^+^).

#### 3.4.4. Synthesis of Compound **4**

Compound **3** (80.8 g, 0.191 mol) was dissolved in DCM (520 mL). Then, 85% phosphoric acid solution (91.7 mL) was added slowly and stirred overnight. Then, 1850 mL of 10% sodium carbonate aqueous solution was added to the reaction system and stirred, from which white solid formed. A total of 49.0 g (0.152 mol) of white solid was obtained after filtering, washing and drying with a yield of 79.6%. 

#### 3.4.5. Synthesis of 221s (2,9)

Compound **4**, HOBT(20.6 g, 0.152 mol), and EDCI(72.8 g, 0.380 mol) were dissolved in 1100 mL DMF. Then, tanshinol-DMF solution was dropped slowly into an ice bath and stirred overnight. Then, saturated sodium chloride aqueous solution was added, after which the reaction mixture was extracted by ethyl acetate 5 times and washed by 5% sodium bicarbonate aqueous solution and water. Finally, 43.2 g of white solid (221s (2,9), 0.086 mol) was obtained with a yield of 59.7% after drying the products using anhydrous sodium sulfate, condensing and vacuum-drying.

^1^H NMR (400 MHz, CDCl_3_) δ 8.03 (s, 1H), 7.52 (d, *J* = 8.0 Hz, 1H), 6.80–6.70 (m, 2H), 6.61 (dd, *J* = 8.2, 2.0 Hz, 1H), 6.36 (s, 1H), 4.92 (dt, *J* = 10.1, 2.4 Hz, 1H), 4.74 (p, *J* = 7.1 Hz, 1H), 4.26 (dd, *J* = 8.6, 4.2 Hz, 2H), 3.83–3.76 (m, 1H), 3.73 (dd, *J* = 12.3, 5.1 Hz, 1H), 3.60 (dt, *J* = 10.0, 6.4 Hz, 1H), 3.09 (dd, *J* = 14.1, 3.6 Hz, 1H), 2.81 (dd, *J* = 14.1, 8.7 Hz, 1H), 2.32 (tt, *J* = 10.1, 3.9 Hz, 1H), 2.19 (dq, *J* = 12.2, 8.0 Hz, 1H), 2.03 (p, *J* = 6.8 Hz, 2H), 1.94 (dt, *J* = 11.4, 4.9 Hz, 1H), 1.85 (ddd, *J* = 11.6, 8.4, 4.2 Hz, 1H), 1.73 (dd, *J* = 10.4, 6.0 Hz, 1H), 1.67 (t, *J* = 4.5 Hz, 1H), 1.37 (d, *J* = 6.9 Hz, 3H), 1.31–1.22 (m, 2H), 1.00 (dd, *J* = 13.8, 3.5 Hz, 1H), 0.87 (d, *J* = 7.5 Hz, 6H), 0.79 (s, 3H).

^13^C NMR (101 MHz, CDCl_3_) δ 173.43, 171.90, 171.55, 143.93, 143.64, 121.29, 116.66, 115.17, 81.00, 72.87, 59.36, 48.92, 47.94, 47.11, 46.35, 44.84, 39.93, 36.49, 29.13, 27.99, 27.15, 24.75, 19.64, 18.76, 17.61, 13.49.

Calculate *m*/*z* from C_27_H_38_N_2_O_7_: 502.27, found: 503.27527 ([C_27_H_38_N_2_O_7_ + H]^+^).

### 3.5. Determination of Purity

#### 3.5.1. HPLC

The sample separation was performed with the Agilent 1260 HPLC and TC-C18 column (250 mm × 4.6 mm, 5 μm) received from Agilent Technologies (Santa Clara, CA, USA). The injection volume was 10 μL. As shown in Appendix A, the standards and samples were separated vis gradient elution (Appendix A). The flow rate was 1.0 mL/min, column temperature 30 °C, and detection wavelength 280 nm. The final purity of the compound was confirmed by deducting the weight loss on drying and the residue on ignition.

#### 3.5.2. Thermal Analysis

Q600 Simultaneous DSC-TGA and Q1000 TGA were used for the determination of melting point. The conditions of DSC were as follows: the flowing of N_2_ (50 mL·min^−1^); temperature increasing rate: 5 °C·min^−1^; and scan region: 25~200 °C. The condition of DSC as follows: the flowing of N_2_ (50 mL·min^−1^); temperature increasing rate: 20 °C·min^−1^; and scan region: 25~800 °C.

#### 3.5.3. The Safety Evaluation of 221s (2,9)

The safety Evaluation of 221s (2,9) was evaluated through the oral administration of 221s (2,9).

### 3.6. The Hypotensive Effect and the Underlying Mechanism of 221s (2,9) on RAAS in SHR Rats

#### 3.6.1. Animals and Animal Care

All SHR rats were randomly divided into 4 groups for 4 weeks (8 mice/group): (1) SHR model group, given CMC-Na for 4 weeks; (2) Captopril group, given oral administration of Captopril (30 mg/kg/day) for 4 weeks; (3) high-dose 221s (2,9) group, given oral administration of high dose of 221s (2,9) (30 mg/kg/day) for 4 weeks; (4) low-dose 221s (2,9) group, given oral administration of high dose 221s (2,9) (15 mg/kg/day) for 4 weeks; and WKY rats were used as negative control group.

#### 3.6.2. Measurement of Blood Pressure, Heart Rate and Body Weight

Blood pressure, heart rate and body weight were measured once a week. The caudal arterial pressure was measured by a Kent non-invasive blood pressure measuring instrument after oral administration.

#### 3.6.3. Measurement of REN, AngII, ALD, NO and ET-1 in Serum

At the end of treatment, mice were euthanized 2 min after injection. Blood were collected and centrifuged at 3000 rpm for 10 min to obtain serum. The concentration of serum REN, AngII, ALD, NO and ET-1 were measured by ELISA kit.

#### 3.6.4. Histopathological Study

Myocardium of rats were harvested in each group for further histological including HE staining and Masson staining. Image J was used to calculate the positive area percentage.

#### 3.6.5. Statistical Analysis

All quantitative results are presented as the mean ± standard deviation. GraphPad Prism 6.0 software (GraphPad Software, San Diego, CA, USA) was used for statistical analysis. The data were evaluated using an unpaired two tailed *t* test. ANOVA tests were used to compare two groups among multiple groups. A value of *p* < 0.05 was considered statistically significant.

## 4. Conclusions

In this paper, 221s (2,9), an ACE inhibitor with 99.5% purity, was synthesized based via the research strategy of multi-molecule synergy with combing structure of tanshinol, borneol and the mother nucleus of ACEI.

The compound 221s (2,9) demonstrated a good safety record, and improved hypertension in SHR rats, the mechanism of which may be related to the RAAS regulation and NO production. Compound 221s (2,9) also could protect cardiac function, improve the morphology of cardiac cells and relieve the development of cardiac fibrosis in SHR.

In summary, these findings provide exidence that 221s (2,9) has the potential to become a novel drug with which to provide a therapy for the prevention and treatment of hypertension. The effort to obtain a better understanding of 221s (2,9) will require further investigation is necessary in order to verify the effect of 221s (2,9) in other target tissues and establish its pharmacokinetic parameters.

## Data Availability

Not applicable.

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
