# Peer review of "Chemical Synthesis, Safety and Efficacy of Antihypertensive Candidate Drug 221s (2,9)"

_molecules, 2023, doi:10.3390/molecules28134975_

Round 1
Reviewer 1 Report
Hypertension is a very topical topic. Therefore, I appreciate the effort taken here to present the newly synthesized compound 221s (2,9), which contains tansinol, borneol and the mother nucleus of ACEI 13. The great advantage of this work is the achieved purity of the product, which was over 99%. The application values of this synthesis are very up-to-date. Preparatory work showed a knowledge of the art of organic chemistry. It may be worth considering by the authors to present a potential mechanism. Analytical analysis of relationships at a very good level and comparative.
I have no negative comments on the content. Maybe it's worth to diversify the graphics a bit and add a graphical abstract to emphasize the value of the topic.
Author Response
We have added a graphical abstract.

Reviewer 2 Report
Dear Authors,
This is an original work submitted by Qin et al. entitled: Chemical Synthesis, Safety and Efficacy of Antihypertensive Candidate Drug 221s (2,9).
This manuscript is novel, well constructed and presents sufficient results on the design, synthesis, characterization and safety and efficacy of the antihypertensive drug 221s (2,9) based on tanshinol, borneol and ACEI mother nucleus. The results are relevant as it showed a real candidate as an antihypertensive drug.
Minor revisions are suggested before the publication of the manuscript.
Line 36 ACEI is one of your keywords and as this is the first time you mentioned in the manuscript, so you need to write the meaning of the acronym.
Line 42 What about hypotension? eg. DOI: 10.1136/bmj.286.6368.832
Line 91 Did you explore another alternatives as mild deprotection oxalyl chloride (DOI 10.1039/d0ra04110f)
Line 217 In this scheme it would be interesting to have the yields at each step.
Carefully check some punctuation marks mistakes such as:
Line 12 space “. In”
Lines 47, 55, 63, etc. References before full stop
Author Response
Point 1: Line 36 ACEI is one of your keywords and as this is the first time you mentioned in the manuscript, so you need to write the meaning of the acronym.
Response 1: The meaning for ACEI has been added.
Angiotensin converting enzyme inhibitors (ACEI) can significantly reduce the concentration of Ang II in plasma and inhibit the pressor effect of exogenous Ang I by directly inhibiting ACE and inhibiting the production of angiotensin II in RAAS[8].
Point 2: Line 42 What about hypotension? eg. DOI: 10.1136/bmj.286.6368.832
Response 2: The adverse effect of Hypotension for ACEI was added.
Revision: However, ACEI have various adverse effects such as irritant dry cough[17-19], vascular edema[20], hyperkalemia[21, 22] and renal dysfunction[23], hypotension[24].
Point 3: Line 91 Did you explore another alternatives as mild deprotection oxalyl chloride (DOI 10.1039/d0ra04110f)
Response 3: In the process of removing the protection of tert-butoxyl group, we tried many methods, among which the effect of phosphoric acid was the best and the hydrolysis degree of ester was the lowest. When we used phosphoric acid to remove the amino protection group, the yields were all over 80% , and the reaction conditions were mild. In addition, the toxicity and stability of phosphoric acid are better than oxalyl chloride, more suitable for the late amplification of the reaction process.
Point 4: Line 217 In this scheme it would be interesting to have the yields at each step.
Response 4: The yields were added in scheme 1.
Scheme 1 Synthesis route of 221s (2,9)
Point 5: Carefully check some punctuation marks mistakes such as:
Line 12 space “. In”
Lines 47, 55, 63, etc. References before full stop
Response 5: The punctuation marks mistakes was checked and corrected.

Reviewer 3 Report
1. Abstract must be improved stating the process of synthesis, methodology of characterization and efficacy studies supported with brief results.
2. Introduction should be improved emphasizing on the objective of the research in last paragraph.
3. The procedure for the synthesis of 221s (2,9) (Line 73-77 is confusing). On one hand authors are talking about mother nucleus of ACEIT while on other side proline and alanine, and the two amino acids and danshensu are being discussed as mother nucleus. Please clarify.
4. Line 103-112, how the purity percentage can be 998 without any unit? Furthermore what do you mean by HPLC purity. The whole paragraph is confusing. Please clarify.
5. Line 149-154; There is description, how the levels of NO and endothelin-1 were determined?
6. Line 158-164; Nothing is clear what was the purpose of staining? Also nothing clear, what was being stained?
7. There is no detail about the HPLC method. No detail about the mobile phase and other HPLC parameters etc.
8. Reference citation is wrong in text in may places. Some references are cited after the full stop. Please verify all the references as per mdpi format. Also check references in the list.
English language is very poor and needs extensive improvement. It is necessary to be improved by English language editing service or any native English speaker.
Author Response
Point 1: Abstract must be improved stating the process of synthesis, methodology of characterization and efficacy studies supported with brief results.
Response 1: The synthesis process and characterization methods were added to the abstract.
Revision: In this paper, a novel compound 221s (2,9) which include tanshinol, borneol and mother nucleus of ACEI was synthesized by condensation esterification, deprotection, amidation, deprotection, and amidation with borneol as the initial raw material using the strategy of combinatorial molecular chemistry. The structure of the compound was confirmed by 1HNMR, 13CNMR, and high-resolution mass spectrometry, with a purity of more than 99.5%.
Point 2: Introduction should be improved emphasizing on the objective of the research in last paragraph.
Response 2: The research objectives was added in Introduction
The main objective of this study is to scale up the synthesis and quality evaluation of candidate compound 221s (2,9), with a focus on the acute toxicity, antihypertensive efficacy, and preliminary safety of the compound. This study will promote the preclinical study of 221s (2,9) as a candidate compound.
Point 3: The procedure for the synthesis of 221s (2,9) (Line 73-77 is confusing). On one hand authors are talking about mother nucleus of ACEIT while on other side proline and alanine, and the two amino acids and danshensu are being discussed as mother nucleus. Please clarify.
Response 3: An ACEI Antihypertensive drug with tanshinol, borneol and the mother nucleus of ACEI designed and synthesized based on drug molecular structure design. Proline and alanine in the structure constitute the mother nucleus of ACEI. Danshensu and borneol are added to improve the synergistic antihypertensive effect of drugs, organ protection, and the ability of drugs to pass through the Blood–brain barrier.
Point 4: Line 103-112, how the purity percentage can be 998 without any unit? Furthermore what do you mean by HPLC purity. The whole paragraph is confusing. Please clarify.
Response 4: The factor of spectral purity was 998, which was provided without. HPLC purity refers to chromatographic purity confirmed by HPLC. The whole paragraph has been clarified.
The objective samples were analyzed and calibrated using HPLC method with gradient elution. The factor of spectral purity for the main maximum peak was confirmed to be 998 on the DAD detector, which showed that the main peak only stood for a single component (Figure 2s). The purity was confirmed by HPLC to be more than 99.87%, which was calculated by the area normalization method, and the total impurity content was 0.13% (Figure 3s, Table 2s).
Point 5: Line 149-154; There is description, how the levels of NO and endothelin-1 were determined?
Response 5: The levels of NO and endothelin-1 were listed in section 3.6.3.
Point 6: Line 158-164; Nothing is clear what was the purpose of staining? Also nothing clear, what was being stained?
Response 6: HE staining was using to assess the effect of 221s (2,9) on cardiac histopathological changes in mice. Chromatin and ribophagy was stained as violet-blue and the ingredient of cytoplasm and extracellular matrix was stained as red.
Point 7: There is no detail about the HPLC method. No detail about the mobile phase and other HPLC parameters etc.
Response 7: The HPLC method has been described in detail and provided in Table 1s.
Sample separation was performed with Agilent 1260 HPLC and a TC-C18 column (250 mm*4.6 mm, 5μm) from Agilent Technologies. As shown in Table 1s, the standards and samples were separated by gradient elution(Table 1s). Flow rate is 1.0 mL/min, column temperature 30℃, detection wavelength 280 nm. The final purity of the compound was confirmed by deducting the weight loss on drying and the
Point 8: Reference citation is wrong in text in may places. Some references are cited after the full stop. Please verify all the references as per mdpi format. Also check references in the list.
Response 8: The order of citations has been corrected one by one, and all citations have been marked before the full stop.
residue on ignition.

Round 2
Reviewer 3 Report
Accept with minor improvement in English language
Little bit improvement in English language is required.